# A Multiparametric MR-Based RadioFusionOmics Model with Robust Capabilities of Differentiating Glioblastoma Multiforme from Solitary Brain Metastasis

**DOI:** 10.3390/cancers13225793

**Published:** 2021-11-18

**Authors:** Jialiang Wu, Fangrong Liang, Ruili Wei, Shengsheng Lai, Xiaofei Lv, Shiwei Luo, Zhe Wu, Huixian Chen, Wanli Zhang, Xiangling Zeng, Xianghua Ye, Yong Wu, Xinhua Wei, Xinqing Jiang, Xin Zhen, Ruimeng Yang

**Affiliations:** 1Department of Radiology, the Second Affiliated Hospital, School of Medicine, South China University of Technology, Guangzhou 510180, China; wujl@hku-szh.org (J.W.); mcweiruili0825@mail.scut.edu.cn (R.W.); mclswroseway@mail.scut.edu.cn (S.L.); zhezhe0301@st.btbu.edu.cn (Z.W.); eychx@scut.edu.cn (H.C.); wanlizhang@st.btbu.edu.cn (W.Z.); eyxinhuawei@scut.edu.cn (X.W.); eyjiangxq@scut.edu.cn (X.J.); 2Department of Radiology, The University of Hong Kong Shenzhen Hospital, Shenzhen 518000, China; 3School of Biomedical Engineering, Southern Medical University, Guangzhou 510515, China; lfr19971014@i.smu.edu.cn; 4School of Medical Equipment, Guangdong Food and Drug Vocational College, Guangzhou 510520, China; laiss@gdyzy.edu.cn; 5State Key Laboratory of Oncology in South China, Sun Yat-sen University Cancer Center, Collaborative Innovation Center for Cancer Medicine, Guangzhou 510060, China; lvxf@sysucc.org.cn; 6Department of Medical Imaging, Sun Yat-sen University Cancer Center, Collaborative Innovation Center for Cancer Medicine, Guangzhou 510060, China; 7Department of Radiology, Huizhou Municipal Central Hospital, Huizhou 516001, China; 202020157793@mail.scut.edu.cn; 8Department of Radiation Oncology, 1st Affiliated Hospital, Zhejiang University, Hangzhou 310009, China; hye1982@zju.edu.cn; 9Department of Oncology, the Second Affiliated Hospital, School of Medicine, South China University of Technology, Guangzhou 510180, China; eywuyong@scut.edu.cn

**Keywords:** glioblastoma multiforme, solitary brain metastasis, MRI, radiomics, fusion

## Abstract

**Simple Summary:**

Glioblastoma multiforme (GBM) and solitary brain metastasis (SBM) are common brain tumors in adults. The two tumors often pose a diagnostic dilemma owing to their similar features on conventional magnetic resonance imaging (MRI). Ability to discriminate the two tumors is critical as it informs clinical treatment strategies. This pilot study attempts to employ the machine learning technique to identify GBM and SBM by fusing radiomics features of multiple MRI sequences and multiple models. A multiparametric MR-based RadioFusionOmics (RFO) model was developed and has demonstrated promising prediction accuracy for the identifications of GBM and SBM.

**Abstract:**

This study aimed to evaluate the diagnostic potential of a novel RFO model in differentiating GBM and SBM with multiparametric MR sequences collected from 244 (131 GBM and 113 SBM) patients. Three basic volume of interests (VOIs) were delineated on the conventional axial MR images (T_1_WI, T_2_WI, T_2__FLAIR, and CE_T_1_WI), including volumetric non-enhanced tumor (nET), enhanced tumor (ET), and peritumoral edema (pTE). Using the RFO model, radiomics features extracted from different multiparametric MRI sequence(s) and VOI(s) were fused and the best sequence and VOI, or possible combinations, were determined. A multi-disciplinary team (MDT)-like fusion was performed to integrate predictions from the high-performing models for the final discrimination of GBM vs. SBM. Image features extracted from the volumetric ET (VOI_ET_) had dominant predictive performances over features from other VOI combinations. Fusion of VOI_ET_ features from the T_1_WI and T_2__FLAIR sequences via the RFO model achieved a discrimination accuracy of AUC = 0.925, accuracy = 0.855, sensitivity = 0.856, and specificity = 0.853, on the independent testing cohort 1, and AUC = 0.859, accuracy = 0.836, sensitivity = 0.708, and specificity = 0.919 on the independent testing cohort 2, which significantly outperformed three experienced radiologists (*p* = 0.03, 0.01, 0.02, and 0.01, and *p* = 0.02, 0.01, 0.45, and 0.02, respectively) and the MDT-decision result of three experienced experts (*p* = 0.03, 0.02, 0.03, and 0.02, and *p* = 0.03, 0.02, 0.44, and 0.03, respectively).

## 1. Introduction

Primary glioblastoma multiforme (GBM; WHO grade IV glioma) and intracranial metastases are commonly identified malignant brain tumors in adults [1]. Noninvasive discrimination between the two tumors helps in the selection of appropriate treatment options and clinical management strategies [2,3]. However, in cases of solitary brain metastasis (SBM), differentiating the GBM from SBM solely based on conventional characteristics on magnetic resonance imaging (MRI) remains challenging. GBM and SBM often share similar radiological manifestations, such as an intratumoral necrotic center and heterogeneous enhancement of the component surrounded by peritumoral edema regions [4,5]. 

To define a better diagnostic tool, previous studies have explored various advanced quantitative MR techniques, such as diffusion weighted imaging (DWI), diffusion tensor imaging (DTI), dynamic contrast-enhanced (DCE), dynamic susceptibility contrast (DSC) perfusion, arterial spin labeling (ASL) perfusion, and spectroscopy, aimed towards excavating more hidden imaging information that would help characterize the physiological and metabolic differences between GBM and SBM [6,7,8,9,10,11,12]. Whereas the data were inspiring, their attempts failed to provide adequate diagnostic confidence due to the inconsistencies in the MR-derived parameters. Besides, these alternatives were associated with extra expense and more scanning time required for implementation, which impeded the application of these advanced MR techniques in routine diagnostic practices.

The advent of radiomics in recent years has tremendously permeated clinical applications in terms of diagnosis, prognosis, as well as prediction of treatment response in tumors [13,14]. In particular, many studies have employed radiomics for the differentiation of GBM from SBM based on conventional MR sequences or/and more advanced MR technologies such as DWI, DTI, APT [15,16,17,18,19,20]. These investigations have shed light on mining more concealed image textures barely captured by the naked eyes in distinguishing GBM from SBM. Nevertheless, these techniques are limited by the fact that the tumor VOI under investigation is tentatively chosen from either the peritumoral edema region [12,15,21] or the enhanced tumor area [4,11]. In addition, there is no definitive data on the tumor VOI (necrotic centers, enhancing margins, or peritumoral edema) that better characterizes GBM and SBM. Besides, radiomics analysis was performed on a single MR sequence, such as contrast-enhanced T_1_WI [22,23], ADC [20], DTI [17], or multi-sequences [18,19,24]. The MR sequence selection is often random and heuristic, and superiority among the MR sequences remains unclear. In fact, the different MR sequences can be viewed as multimodality images that depict the tumor from different perspectives. Fusion of imaging information from multiple MR sequences gives better characterization of the tumor properties and would theoretically enhance the classification performance of a radiomics model. To date, no attempts have been made to fully exploit the potential of integrating radiomics features acquired from multiple MR sequences for the discrimination of GBM from SBM. In addition, the reported radiomics models are often built on a single classifier. A model’s classification capability is linked with the classifier used. Indeed, different classifiers might yield inconsistent results even when applied on the same task [25]. An ensemble of classifiers, a process analogous to disease diagnosis by a multi-disciplinary team (MDT), usually offers more reliable and reproducible performance compared to a single classifier [15,26].

Here, we hypothesized that integrating information from multiple MR sequences and various classifiers could yield a more robust radiomics system. We proposed and used a novel hierarchical fusion radiomics model, referred to as RadioFusionOmics (RFO), to fuse multimodality radiomics features from multiple MR sequences, and combine different classifier outputs via an MDT-like fusion method. The technique was comprehensively tested with optimal lesion VOI and MR sequence combinations. In addition, the technique competed with human expert diagnosis and yielded top features that could differentiate between GBM and SBM.

## 2. Materials and Methods

### 2.1. Patient Cohort

This study was approved by the Institutional Review Board of Guangzhou First People’s Hospital, Huizhou Municipal Central Hospital, and Sun Yat-sen University Cancer Center. The requirement for informed consent from individual patients was waived due to the retrospective nature of the study design. A total of 458 patients with histopathologically confirmed GBMs or SBMs, who underwent diagnostic preoperative MRI from September 2007 to September 2021, were evaluated for recruitment in the study. Patients who received treatments, including chemotherapy, radiotherapy, or surgery, before MRI examination, those who lacked preoperative MRI images or the presence of artifacts in the MRI images, making lesion delineation difficult, and those with multiple lesions (multicentric/multifocal glioblastoma, multiple metastases) were excluded from the study. Patients with a maximum lesion diameter < 10 mm, those with a lesion with a pure solid or cystic component, or without enhancement, as well as those with a lesion with skull invasion were also excluded. Ultimately, a total of 244 patients (131 GBMs and 113 SBMs) were enrolled (Figure 1). The included patients were then divided into a training/validation cohort (GBM, *n* = 61; SBM, *n* = 60), an independent testing cohort 1(GBM, *n* = 33; SBM, *n* = 29), and an independent testing cohort 2(GBM, *n* = 37; SBM, *n* = 24). The 113 SBM patients included 44 cases with primary lung cancer, 7 with primary breast cancer, 4 with primary esophageal cancer, 7 with primary colorectal cancer, 1 with primary prostate cancer, 1 with primary ovarian cancer, 1 with primary endometrial carcinoma, 1 with primary type B3 thymoma, 1 with primary malignant mixed tumor of salivary gland, 1 with primary nasopharyngeal carcinoma, and 45 with a cancer of unknown origin. All the patients’ records and medical information were anonymized and de-identified prior to analysis. 

### 2.2. Image Acquisition and Histopathology

All MRI examinations were performed on a 3.0 T MR scanner (Verio, Siemens, Erlangen, Germany) and a 1.5 T MR scanner (Signa EXCITE HD, GE Healthcare, Milwaukee, WI, USA), equipped with a 12-channel head coil. The following sequences were acquired: axial T_1_-weighted image (T_1_WI) (repetition time/echo time (TR/TE), 400~2000/9~15 ms; section thickness, 6 mm; number of signals acquired, 2; matrix, 320 × 320; field of view (FOV), 230 × 230 mm), axial T_2_-weighted image (T_2_WI) ((TR/TE, 3500~6000/95~100 ms; section thickness, 6 mm; number of signals acquired, 1; matrix, 320 × 320; FOV, 230 × 230 mm), and axial T_2_-weighted fluid attenuated inversion recovery (T_2__FLAIR) images ((TR/TE, 6000~8000/90~120 ms; section thickness, 6 mm; number of signals acquired, 1; matrix, 320 × 320; FOV, 230 × 230 mm)). In addition, axial, sagittal, and coronal post-contrast T_1_WIs were acquired after intravenous administration of 0.1 mmol/kg gadolinium diethylenetriamine penta-acetic acid (Bayer Healthcare, Berlin, Germany).

### 2.3. Delineation of Volume of Interest (VOI) 

All the images were stored in the Digital Imaging and Communications in Medicine (DICOM) file format. The three-dimensional VOI of the tumors were manually delineated, slice-by-slice, using ITK-SNAP software (http://www.itksnap.org; assessed on 15 January 2020) on the axial plane on the T_2__FLAIR images. The axial T_1_WI, T_2_WI, and post-contrast axial T_1_WI were used to cross-check the extent of the tumor or peritumor edema and to fine-tune the contour. The exclusion criteria resulted in a patient cohort with comparable MR images presenting similar pathology-related components (i.e., intratumoral necrotic center, enhancing component, and peritumoral edema that simultaneously presented in GBMs or SBMs). Specifically, visible tumor margins were manually delineated to obtain the volume of non-enhanced tumor (nET), enhanced tumor (ET), and peritumoral edema (pTE) (Figure 2). The procedure was conducted by two investigators (J. Wu and R.M Yang, with 4 and 15 years of experience in radiological diagnosis, respectively). The conformity of the delineated VOIs were evaluated by the Dice similarity coefficient. Intersection of two VOIs was used as the final VOI based on whether the Dice index was greater than 0.9. Discrepancies on the lesion boundary (Dice < 0.9) were resolved by further discussions until there was mutual consensus. Besides, morphological union operation was carried to combine two/three of the three VOIs, which resulted in a total of seven VOIs (VOI_nET_, VOI_ET_, VOI_pTE_, VOI_nET+ET_, VOI_nET+pTE_, VOI_ET+pTE_, and VOI_nET+ET+pTE_).

### 2.4. Extraction of Radiomics Features

Imaging normalization was performed on all the MRI images before feature extraction. A total of 109 radiomics features were extracted on each of the seven VOIs from all the four MRI sequences (T_1_WI, CE_T_1_WI, T_2_WI, and T_2__FLAIR) using the open-source radiomics toolkit Pyradiomics (https://pyradiomics.readthedocs.io/en/latest/index.html; accessed on 15 January 2020). The extracted features included first order features (*n* = 19), shape features (*n* = 15), gray level co-occurrence matrix (GLCM) features (*n* = 24), gray level size zone matrix (GLSZM) features (*n* = 16), gray level run length matrix (GLRLM) features (*n* = 16), neighboring gray tone difference matrix (NGTDM) features (*n* = 5), and gray level dependence matrix (GLDM) features (*n* = 14) [27].

### 2.5. RadioFusionOmics 

The proposed RadioFusionOmics (RFO) methodology is a novel two-level fusion scheme that integrates radiomics information from different MRI sequences (Figure 3). It exploits the ensemble learning to fully explore the strengths of various classifiers.

#### 2.5.1. Level 1: Feature Fusion

We proposed a novel feature fusion method that integrates features extracted from different MRI sequences (Figure 3b). We used a data matrix $X_{\left(p\times n\right)}$ for a particular feature, where p (p=2, 3 or 4) denotes the number of MRI sequences, and *n* is the training sample size. To incorporate the class structure (i.e., memberships of the training samples in class) into the feature fusion, a between-class scatter matrix $S=\sum_{i=1}^{c}n_{i}\left({\bar{x}}_{i}-\bar{x}\right){\left({\bar{x}}_{i}-\bar{x}\right)}^{T}$ was constructed by dividing matrix X into c separate groups (c is the number of classes, c=2); where $n_{i}$ columns belong to the *i*th class ($n=\sum_{i=1}^{c}n_{i}$), and the ${\bar{x}}_{i}$ and $\bar{x}$ represent the means of the $x_{i,j}$ vectors (*j*^th^ sample in the *i*^th^ class) in the *i*th class and the whole set. Unitizing the between-class matrix S generated a transformation matrix W, which was used to reduce the dimensionality of matrix X.

Feature fusion was accomplished by mapping $W^{T}X$ to compress matrix X to a row vector $x_{\left(1\times n\right)}$ (see Appendix B for algorithm details). The fusion was repeated for all the *m* (*m* = 109) features and stacking the *m* row vectors $x_{\left(1\times n\right)}$ yielded the fused feature matrix $F_{\left(m\times n\right)}$. Unlike the serial or parallel fusion, the proposed method not only fuses the features from different MR sequences but also eliminate the between-class correlations and restricts the within-class correlations during the feature fusion process [28]. More technical details can be found in Appendix B.

With features $F_{seq}^{1}$, $F_{seq}^{2}$,$\;F_{seq}^{3}$, and $F_{seq}^{4}$, respectively, extracted from the four MRI sequences—T_1_WI, CE_T_1_WI, T_2_WI, and T_2__FLAIR—the feature level fusion was done by either combining any two ($F_{seq}^{1;2}$, $F_{seq}^{1;3}$,$\;F_{seq}^{1;4}$, $F_{seq}^{2;3}$, $F_{seq}^{2;4}$, and $F_{seq}^{3;4}$), three ($F_{seq}^{1;2;3}$, $F_{seq}^{1;2;4}$, $F_{seq}^{1;3;4}$, and $F_{seq}^{2;3;4}$), or all the four feature vectors ($F_{seq}^{1;2;3;4}$). 

In the training stage, feature level fusion would yield two outputs: the fused feature of the training cohort and the transformation matrix W.

#### 2.5.2. Level 2: Model Fusion

The previously described fused features were used to train fifteen classification models built with different combinations of five feature selection methods and three classifiers (Appendix A). A stratified five-fold cross-validation was performed on the training cohort to rank all the fifteen models and then the top 3 models were screened out.

In addition, the ensemble learning was applied to fuse the top 3 models via a multi-disciplinary team (MDT)-like fusion method. Each of the models was regarded as a clinical specialist providing a classification prediction$\;c_{i},i=\left\{1,2,3\right\}$. Their decisions were integrated by the weighted fusion (WF) method, which assigns different weights $w_{i}=\frac{acc_{i}}{\sum_{k=1}^{3}acc_{k}}$ to the *i*^th^ models according to their accuracies $acc_{i}.$ The accuracies were averaged over the five-fold cross-validation, and reaches a consensus classification by a linear weighted sum$\;\sum_{i}^{3}w_{i}c_{i}\left(x\right)$.

In the training stage, the model level fusion provided a final consensus classification model by fusion of the screened top three trained models coupled with their associated weights $w_{i}$.

#### 2.5.3. Independent Testing

All the patients were divided into a training cohort (*n* = 121), an independent testing cohort 1 (*n* = 62), and an independent testing cohort 2 (*n* = 61). The transformation matrix W acquired from the feature level fusion in the training stage was directly applied to fuse features of the MRI sequences in the testing cohort. The fused testing features were fed into the final classification model for independent testing.

### 2.6. Evaluation of the Model

#### 2.6.1. Study 1: Comparison of Lesion VOI

We compared the discriminative power of the features extracted from the seven VOIs (VOI_nET_, VOI_ET_, VOI_pTE_, VOI_nET+ET_, VOI_nET+pTE_, VOI_ET+pTE_, and VOI_nET+ET+pTE_) on each MRI sequence. This was used to verify the lesion VOI that could provide better discrimination between the GBM and SBM.

#### 2.6.2. Study 2: Comparisons of the Different Combinations of Mri Sequences Used in the Fusion

Using the best lesion VOI, different MRI sequences were fused via the proposed methodology; $F_{seq}^{1;2}$, $F_{seq}^{1;3}$, …, $F_{seq}^{1;2;3;4}$). We then compared the various combinations of MRI sequences for fusion.

#### 2.6.3. Study 3: Comparison with Radiologist Performance

The constructed RFO model was finally built with features from the best lesion VOI through the best combination of MRI sequences that resulted from Study 1 and 2. The performance was measured by the area under the receiver operating characteristic (ROC) curve (AUC), accuracy (ACC), sensitivity (SEN), and specificity (SPE), and were compared with human diagnosis by three radiologists (W.L Zhang, J.L Wu, and R.M Yang, with 3, 5, and 15 years of experience in neuroradiology, respectively), without prior knowledge of the patients’ pathology. For further illustration, the performance of our proposed RFO model was also compared with the clinical diagnosis result by a multi-disciplinary team (MDT), which was composed by three specialists (R.M Yang with 15 years of experience in radiology, X.H Ye with 15 years of experience in radiation oncology, and Y Wu with 19 years of experience in oncology). By counting the number of decisions made by the three specialists in each class, the class with the highest number of votes was regarded as the result of MDT.

#### 2.6.4. Study 4: Top Features

The top-ranked features associated with the discrimination of GBM or SBM were also sieved by the proposed RFO model and their discriminative capabilities were analyzed. 

### 2.7. Statistical Analysis

All statistical analyses were performed in SPSS 21.0 software (SPSS, Inc.). Continuous variables were reported as the means ± standard deviation (SD) while categorical variables were presented as numbers and proportions. Normality of the data distribution was evaluated by the Shapiro–Wilk test. The demographic characteristics in the GBMs and SBMs were compared using the chi-square test for categorical variables, Student’s *t* test for normally distributed continuous variables, while the Mann–Whitney U test was used for non-normally distributed continuous variables. In addition, we used the paired samples Wilcoxon signed rank test for the comparisons. A *p*-value < 0.05 was considered to be statistically significant.

## 3. Results

### 3.1. Patient Characteristics

This study included a total of 244 patients (131 GBMs and 113 SBMs). There were no significant sex differences between the GBM and SBM in the training and independent testing cohorts (Table 1). Our data showed that the SBM patients were older (58.18 ± 9.83 years for the SBM cohort vs. 52.22 ± 15.59 years for the GBM cohort, *p* = 0.012) and more likely to be in the infratentorial region (22/113 for the SBM cohort vs. 3/131 for the GBM cohort, *p* = 0.001).

### 3.2. Study 1: Comparison of the Lesion VOIs

For each MRI sequence, we extracted the radiomics features from seven VOIs. Each feature (*n* = 4 × 7 types in total) was independently applied in the fifteen models and evaluated by the five-fold cross-validation. The mean AUC values over for all the models and folds were reported (Figure 4). In all the four MRI sequences, image features from the VOI_ET_ were the most superior compared with those from VOI_ET_, VOI_nET_, and VOI_pTE_ and their possible combinations. The best performance in terms of the VOI_ET_ features was seen in the T_2__FLAIR (AUC = 0.92), followed by CE_T_1_WI (0.90), T_2_WI (0.88), and then T_1_WI (0.87). Furthermore, the VOI combinations did not provide significant aid in enhancing the model’s discriminative capability. Even though it was still inferior to VOI_ET_, the VOI_nET+ET_ combination showed better performance on all the MRI sequence except for the CE_T_1_WI. This data might suggest that the enhanced tumor region in MRI provided the most essential diagnostic information in characterizing and differentiating GBM vs. SBM. 

### 3.3. Study 2: Combination of the MRI Sequences for Fusion

Using the best lesion VOI (VOI_ET_), the features of the VOI_ET_ from the four MRI sequences ($F_{seq}^{1}$, $F_{seq}^{2}$,$\;F_{seq}^{3}$, and $F_{seq}^{4}$) were fused by combining any two, three, or all the four sequences via the proposed methodology, and their classification performance was compared (Figure 5). Our data showed that, without feature fusion, the best performance (mean AUC) was observed in T_2__FLAIR ($F_{seq}^{4}$, AUC = 0.94), followed by CE_T_1_WI ($F_{seq}^{2}$, AUC = 0.92), T_2_WI ($F_{seq}^{3}$, AUC = 0.89), and T_1_WI ($F_{seq}^{1}$, AUC = 0.89). Compared with the use of features from a single MRI sequence, feature fusion between any two MRI sequences improved the classification accuracy (except for $F_{seq}^{2;4}$ and $F_{seq}^{3;4})$. The best performance was observed in the fusion of T_1_WI and T_2__FLAIR (i.e., $F_{seq}^{1;4}$, AUC = 0.95). Interestingly, all the fusion associated with $F_{seq}^{4}$ (T_2__FLAIR) yielded satisfactory results $(F_{seq}^{1;4}$, $F_{seq}^{2;4}$,$\;F_{seq}^{3;4}$,$\;F_{seq}^{1;2;4}$,$\;F_{seq}^{1;3;4}$,$\;F_{seq}^{2;3;4}$, and $F_{seq}^{1;2;3;4}$, AUC > 0.93). However, incorporation of more features from other MRI sequences (e.g., $F_{seq}^{1;4}$ vs. $F_{seq}^{1;2;4}$, or $F_{seq}^{1;4}$ vs. $F_{seq}^{1;2;3;4}$) did not necessarily further enhance the accuracy of the fusion. Nevertheless, feature fusion generally outperformed the use of features of a single MRI sequence (mean AUC 0.91 vs. 0.93, *p* = 0.007). The detailed statistical differences between these feature types were also reported (see Appendix A in Appendix A). Thus, $F_{seq}^{1;4}$ emerged as the best feature fusion combination and was used for subsequent RFO modeling.

### 3.4. Study 3: The RFO Model vs. Radiologist Performance

Features extracted from the T_1_WI and T_2__FLAIR sequences on VOI_ET_ were fused to yield the $F_{seq}^{1;4}$ feature level fusion in the RFO model. The fusion was used to train 15 classification models. The top 3 models were singled out and then integrated in the model level fusion to define a consensus model, which was validated using the independent testing cohorts. For comparison purposes, the three top models were individually tested against the two testing cohorts, and their mean AUCs were 91.6% and 86.4%, respectively (Table 2). The data showed that the proposed RFO model improved the classification accuracy to an AUC 92.5% (*p* = 0.04), ACC 85.5% (*p* = 0.10), and SPE 85.3% (*p* = 0.02) on the independent testing cohort 1, while the ACC and SPE on the independent testing cohort 2 were also increased to 83.6% (*p* = 0.04) and 91.9% (*p* = 0.03). For almost all outcomes (except for the SPE of the testing cohort 2), the proposed RFO model significantly preceded all the data from the three board-certified neuroradiologists, who obtained the best performance with AUC 75.4%, ACC 75.8%, SEN 75.8%, and SPE 75.9% on testing cohort 1 and AUC 78.2%, ACC 77.0%, SEN 83.3%, and SPE 73.0% on testing cohort 2 (Table 2). Furthermore, it was interesting to compare our results with the MTD discrimination to mimic daily clinical practice. Our results that the RFO performance was on average about 15% higher than MDT-decision voted by three specialists (one radiologist, one radiation oncologist, and one medical oncologist) further demonstrated the superiority of RFO (Table 2). The statistical differences between the data obtained by the proposed RFO model and the expert diagnosis are shown in Table 3. 

In addition, we evaluated the classification performance following the addition of two statistically significant clinical characteristics (“lesion location” or/and “age”) to the radiomics features in the RFO model (Table 2). The data demonstrated that adding a patient’s clinical characteristics showed different discrimination potentials on two independent test cohorts, as revealed by the statistical comparisons shown in Figure 6. Taking the performance on testing cohort 1 as an example, addition of patient’s age into RFO would decrease the AUC to 92.2% (*p* = 0.01), while addition of the lesion location would significantly improve the classification accuracy to AUC 92.9% (*p* = 0.01), ACC 87.1% (*p* = 0.01), and SEN 89.3% (*p* = 0.00). Interestingly, addition of both clinical characteristics did not further enhance the performance (AUC 92.7%, *p* = 0.02). In contrast, adding age improved the AUC on testing cohort 2 to 86.6% (*p* = 0.00), while there was no performance improvement after location was added (AUC 85.8%, *p* = 0.05). It was worthy to stress that adding both clinical characteristics has a significant performance improvement of AUC 86.5% (*p* = 0.03), ACC 85.2% (*p* = 0.03), and SEN 75.0% (*p* = 0.00). 

### 3.5. Study 4: Highly Correlated Radiomics Markers

The high correlation features (extracted on VOI_ET_) associated with the discrimination of GBM and SBM discrimination also can be selected by the fifteen classification models within the RFO framework. Since each of the 15 models was equipped with a feature selection procedure, we counted and ranked the occurrence of each selected feature (only in models with AUC > 0.8). The ten most frequently selected features included four first-order-based features and six shape-based features (Table 4). Except for the ‘Median’ in T_1_WI, all the features showed statistically significant differences between GBM and SBM. To differentiate between GBM and SBM, the mean feature values of the two groups (i.e., “M” in Table 4) were calculated and used as the threshold. The discrimination capabilities of the top 10 features in T_2__FLAIR were superior to their counterparts in T_1_WI. Three first-order features (‘90th percentile’, ‘Median’, and ‘Maximum’) in T_2__FLAIR, as well as one shape feature (‘Maximum 2D DiameterColumn’) in both T_1_WI and T_2__FLAIR, demonstrated satisfactory discriminative capabilities. The data showed that ~70% of the GBM group had larger feature values, while ~ 70% of the SBM group had smaller values. 

The fused feature of T_1_WI and T_2__FLAIR (i.e., $F_{seq}^{1;4})$ via the proposed RFO model (generated in the feature level fusion) was shown to further enhance the discriminative power in differentiating GBM vs. SBM, especially when compared with the corresponding features from either T_1_WI or T_2__FLAIR (Figure 7).

## 4. Discussion

Accurate distinction between GBM and SBM is critical for accurate radiological assessment, which paves the way for precision medicine and personalized treatment options [29]. Our strict exclusion criteria (excluding the “pure solid or cystic”) left us a “clean” patient cohort of GBMs and SBMs both presenting an intratumoral necrotic center and heterogeneous enhancing component surrounded by peritumoral edema regions on MR. Such cases exhibit the most common imaging manifestations encountered in our daily practice, posing the biggest challenge for radiologists. The proposed RFO model fusing image information from multiple MR sequences was crafted to deal with these tricky and tangled cases that may confound the radiologists in clinics. The RFO model demonstrated satisfactory and superior discrimination capability compared with the experienced radiologists and MDT decision. In addition, we explored the discriminative capabilities of the radiomics features from different tumor volumetric components on different MRI sequences.

Radiomics convert digital medical images into mineable high-dimensional data and have been widely recognized as a practical alternative for noninvasive diagnosis/prognosis in oncology [13,30]. Previous studies demonstrated the feasibility of applying radiomics models to differentiate GBM from SBM based on either conventional MR images or advanced MR technologies [4,17,20,22,31]. Notably, one crucial and defining step for radiomics modeling is the delineation of the lesion VOI. The lesion VOI defines the image context associated with tumor pathological heterogeneity and exerts a direct impact on the extracted radiomics features. Tumor heterogeneity from different VOI components has been associated with poor survival rates in GBM [16,32]. Besides, a radiomics model’s predictive performance also depends on the image features as previously demonstrated [15,21,33]. The previous studies did not, however, resolve the dilemma of whether to derive the lesion VOI from the peritumoral edema region [12,15,21], or from the enhanced tumor area as a whole [19,34]. Evaluation of the features such as necrotic centers, enhancing margins, or peritumoral edema would lead to a more precise and reproducible discriminative model. Indeed, previous reports noted the positive role played by the peritumoral region based on pathological distinctions, such as vasogenic edema, in SBM against a combination with scattered neoplastic cells in GBM [8]. Besides, the infiltrative characteristics of tumor cells in GBMs were also found to exhibit different diffusion, spectroscopic, and perfusion features compared to the peritumoral edema in SBM [8,11]. In contrast, our results showed that features from the VOI_ET_ (enhancing tumor) yielded the most discriminative capability. This observation aligns with a previous study that employed a different imaging (APT) technology [16]. We speculate that the microscopic tumor cell infiltration characteristics are mostly exhibited in the enhancing tumor component, and macroscopically present as intra-tumoral heterogeneity reflected in image texture differences between GBM and SBM. 

On the other hand, selection of an appropriate MR sequence presents another challenge in MR-based radiomics modeling. Previous analyses were mostly performed on a single MR sequence, such as CE_T_1_WI and T_2_WI [19,22,23,34]. However, such preferences were presumably random because no selection criteria were provided. Choosing a proper MR sequence for radiomics modeling strengthens comprehensive comparative studies. For example, Tateishi et al. compared the CE_T_1_WI, T_2_WI, and ADC sequences and showed that the intratumor textures from T_2_WI produced the highest AUC (0.78) in differentiating GBM from SBM [19]. However, the study excluded the T_2__FLAIR sequences in the analysis. In our study, we extensively compared the four conventional MR sequences (T_1_WI, T_2_WI, T_2__FLAIR, and CE_T_1_WI) and their possible combinations and demonstrate the positive role of the T_2__FLAIR sequences in GBM and SBM discrimination.

In addition, we hypothesized that managing the MR sequences in a collaborative way would offer more multifaceted imaging information that is usually absent in a single MR sequence. The novel feature fusion method proposed in the RFO framework performed effective feature fusion through incorporating class structure information, which unitized the constructed between-class matrix in the fusion process. Thus, features from different MR sequences were not only integrated, but the fused features were more representative and discriminative between the GBM and SBM. With this novel feature fusion, we showed that the fused features associated with the T_2__FLAIR sequences (T_1_WI + T_2__FLAIR) demonstrated superior discriminative capabilities as compared with other MR sequences. The best performance in terms of the VOI_ET_ features was in T_2__FLAIR (AUC = 0.92) followed by CE_T_1_WI (0.90) and T_1_WI (0.87). However, feature fusion of ‘T_2__FLAIR+ CE_T_1_WI’ did not outperform feature fusion of ‘T_2__FLAIR+ T_1_WI’, as we expected. This phenomenon was associated with the fact that the image features from the T_2__FLAIR and CE_T_1_WI sequences are universally similar and somewhat overlapping, as opposed to T_2__FLAIR and T_1_WI where the image features might be complementary. 

Furthermore, the novel MDT-like model fusion in the proposed RFO framework was able to mimic the multi-disciplinary decision-making process. The fusion treated the top 3 models as three independent experts, whose opinions were weighted by their experiences (performances in five-fold cross-validations of the training/validation dataset) and in combination to yield a final consensus prediction. This weighted fusion approach outperformed the majority voting method that simply counts the number of decisions from each specialist and then predicts based on the highest number of votes [15] (see Appendix A in Appendix A). 

Our data showed that four first-order-based features and six shape-based features of the VOI_ET_-specific T_1_WI and T_2__FALIR were the top 10 features associated with the discrimination of the GBM vs. SBM. In particular, ‘90th percentile’, ‘Median’, and ‘Maximum’ in T_2__FLAIR, and ‘Maximum 2D DiameterColumn’ in both T_1_WI and T_2__FLAIR exhibited satisfactory diagnostic capabilities [35]. A higher percentile might imply high tumor heterogeneity. For example, Tozer et al. underscored the role of the 90th percentile in the identification of glioma subtypes [36]. Our results showed that the ‘90th percentile’ feature (higher in GBM than in SBM) on the T_2__FLAIR sequence demonstrated the highest discriminative capabilities. This could be ascribed to the fact that GBM is characterized by poorly differentiated neoplastic astrocytes and extensive microvascular proliferation and/or necrosis [37]. Besides, the T_2__FLAIR sequence highlighted the changes in signal intensity within the tumor by suppressing the cerebrospinal fluid signal, thus resulting in a higher ‘90th percentile’ value in GBM. Our results also showed that the ‘Median’ value of GBM was higher than that of SBM. However, whereas its role in distinguishing GBM and SBM was not defined, a previous study had demonstrated its merit in ccRCC and pRCC differentiation [20]. 

The ‘Maximum 2D diameter’ feature, the largest Euclidean distance between the vertices of the tumor surface mesh, is used to describe the tumor size. Our data showed that the mean diameter of GBM was larger than that of the SBM (with ‘M’ value of 6 cm: Table 4). These data were consistent with the radiological and clinical characteristics of the two tumors. The GBM pathology is characterized by angiogenesis and highly infiltrative growth, with enhanced tumor volume. On the contrary, SBM usually exhibits limited expansile growth and tends to compress the surrounding brain tissue with a spherical shape and smaller volume [38]. In addition, SBM usually occurs in the subcortex of the watershed area, which is adjacent to the brain draining veins. The draining veins are easily compressed by the tumor mass effect. Collectively, the factors give rise to the obstruction of venous reflux, leading to obvious peritumoral edema, and triggering of a series of clinical symptoms. Therefore, small size SBM is easier to detect in daily diagnostic practice.

In agreement with the previous observation, the demographics of the enrolled patients showed that SBMs were preferentially located in the infratentorial region compared to GBMs (*p* = 0.001) [31]. In addition, patients with SBM were older than the those with GBM (*p* = 0.001), which is also consistent with previous data [20,23]. Thus, addition of the lesion location to the RFO model can further enhance its performance. 

Nevertheless, our study is limited by the retrospective nature of the design and the fact that the sample size was not large enough due to the strict inclusion criteria. It is worthy to stress that the current enrolled cohort might not necessarily reflect the real distribution of the imaging manifestations in GBMs and SBMs. We found that most of the excluded GBMs (*n* = 11) or SBMs (*n* = 19) manifested a pure solid appearance, and only *n* = 6 GBM cases (but no SBMs cases) presented a pure cystic appearance or without enhancements. Although the RFO model was employed to differentiate the solid GBMs from the solid SBMs, and also exhibited satisfactory performance (see Appendix A in Appendix A), unbalanced case distributions, either universal across different institutions or is specific in our center, might still probably introduce prediction bias and serve as a limitation of a radiomics-based model. Therefore, there is need for further studies with a larger patient population to validate the proposed model’s capability. Besides, our analysis included SBM of different histopathological origins. It is still unknown whether the different origins exhibit different image textures and their possible influences on the model. In addition, an unclear correlation between the radiomics features with pathological manifestations in the enhanced tumor component was also considered a limitation.

## 5. Conclusions

In summary, we successfully built a high-performing RadioFusionOmics model with combined multiple classifiers integrated with multi-sequences feature fusion. Our findings demonstrated that the model outperformed experienced radiologists and might be a promising tool for computer-aided diagnosis of GBM and SBM.

## Figures and Tables

**Figure 1 cancers-13-05793-f001:**
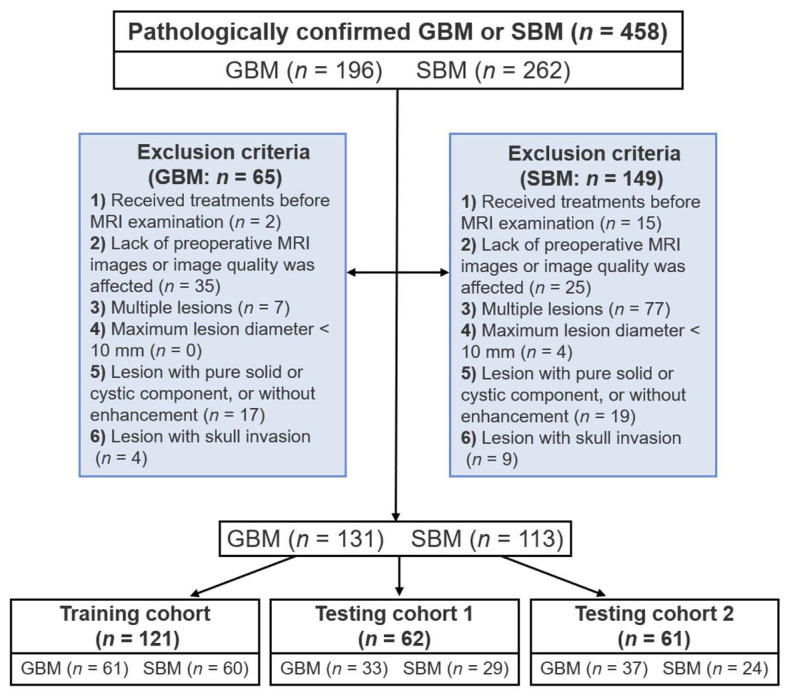
Patients’ inclusion and exclusion criteria.

**Figure 2 cancers-13-05793-f002:**
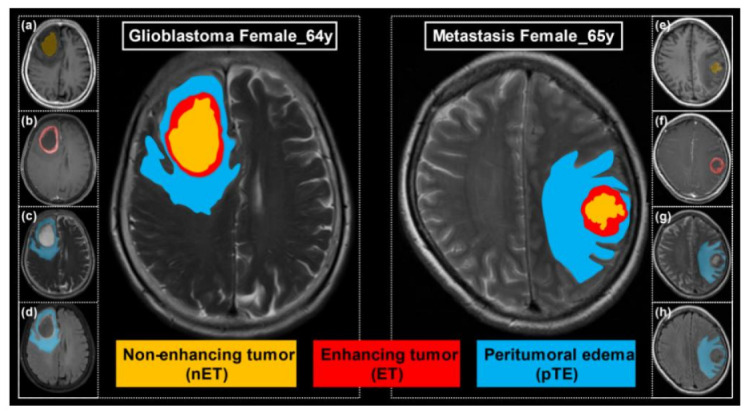
Delineation of non-enhanced tumor (nET), enhanced tumor (ET), and peritumoral edema (pTE); (**a**–**d**) Axial T_1_WI, CE_T_1_WI, T_2_WI, and T_2__FLAIR images of a 64-year-old female with pathologically proven GBM. (**e**–**h**) Axial T_1_WI, CE_ T_1_WI, T_2_WI, and T_2__FLAIR images of a 65-year-old female with pathologically proven SBM. The non-enhancing tumor (nET) and enhancing tumor (ET) are delineated on the CE_T_1_WI and depicted as yellow and red masks, respectively, while the peritumoral edema (pTE) is delineated on the T_2__FLAIR and shown in blue.

**Figure 3 cancers-13-05793-f003:**
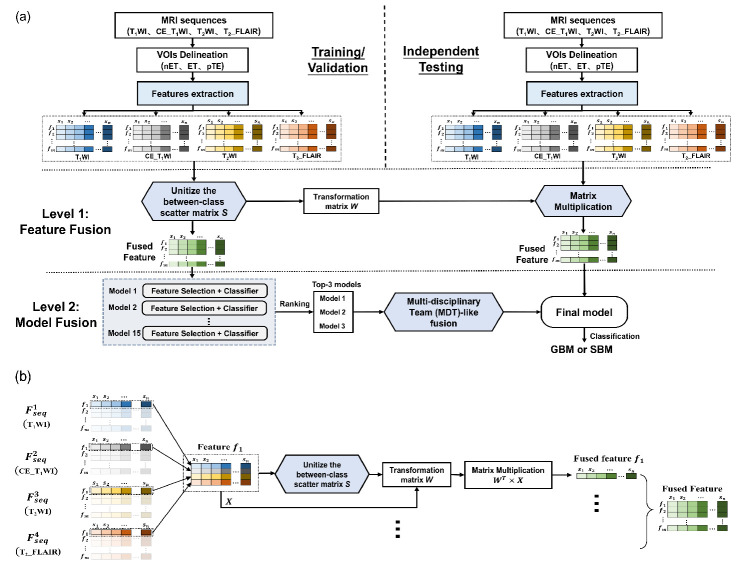
The proposed framework; (**a**) The RadioFusionOmics (RFO) model. (**b**) The “Level 1-feature fusion” in RFO.

**Figure 4 cancers-13-05793-f004:**
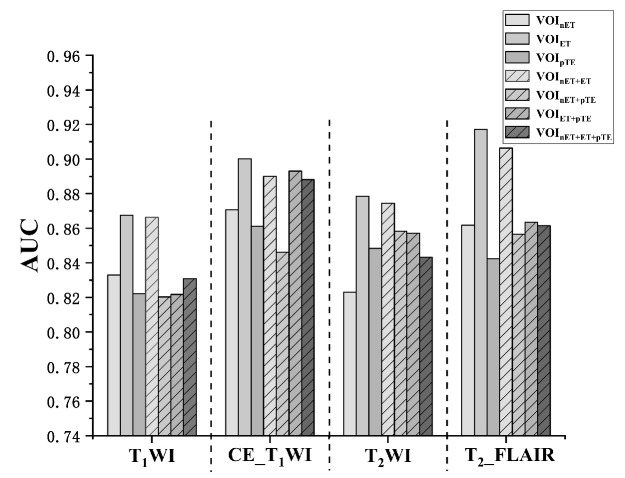
Comparison of the lesion VOIs; Prediction performances of the seven types of ROIs on each image sequence.

**Figure 5 cancers-13-05793-f005:**
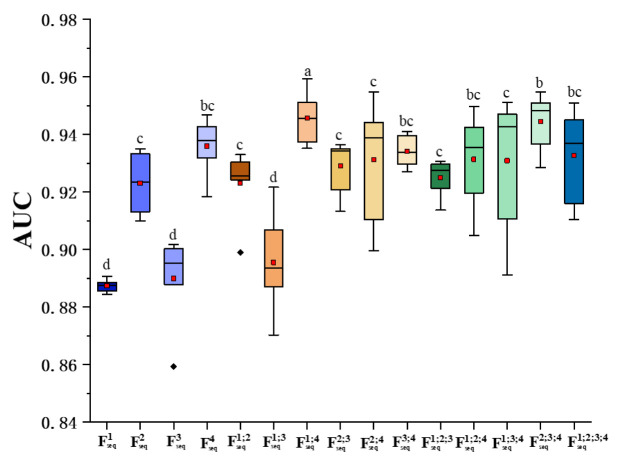
Performance comparisons of different combinations of the MRI Sequences; The $F_{seq}^{num}$ represents the feature extracted from the single sequence num, while the $F_{seq}^{num_{1};num_{2}}$ represents the fused feature of sequence $num_{1}$ and sequence $num_{2}$. Boxplots of the AUC distributions achieved by the 15 discrimination models for each of the 15 feature types. The boxes run from the 25th to 75th percentile; the two ends of the whiskers represent the 5% and 95% percentiles of the data; the horizontal line and the square in the box shows the median and mean values, respectively. The diamonds represent outliers. The letters above each box (i.e., ‘a, b, c, d’) are the symbols to indicate whether a statistically significant difference exists between any two feature types. No common letters indicate that the two feature types are significantly different.

**Figure 6 cancers-13-05793-f006:**
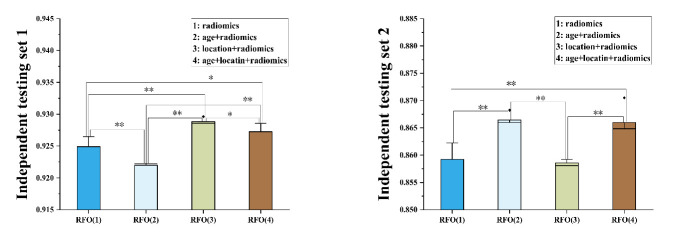
The statistical comparisons of AUC of the RFO model with or without clinical features on two independent testing cohort 1 (left) and 2 (right); Asterisks (** for *p* ≤ 0.01 and * for 0.01 ≤ *p* ≤ 0.05) are marked between any two groups with significant differences.

**Figure 7 cancers-13-05793-f007:**
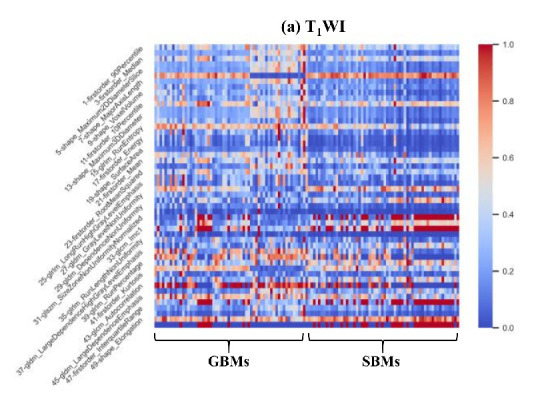
Heat maps of the top 50 features; (**a**) T_1_WI. (**b**) T_2__FLAIR, and (**c**) $F_{seq}^{1;4}$.

**Table 1 cancers-13-05793-t001:** Patient characteristics.

Demographics		Total			Training/Validation Cohort			Independent Testing Cohort 1			Independent Testing Cohort 2		
		GBM (*n* = 131)	SBM (*n* = 113)	*p*-Value	GBM (*n* = 61)	SBM (*n* = 60)	*p*-Value	GBM (*n* = 33)	SBM (*n* = 29)	*p*-Value	GBM (*n* = 37)	SBM (*n* = 24)	*p*-Value
Age, mean ± SD (years)		52.22 ± 15.59	58.18 ± 9.83	0.012 ^c^	49.92 ± 16.07	58.02 ± 9.89	0.001 ^a^	54.79 ± 15.48	59.34 ± 11.29	0.196 ^a^	53.73 ± 14.75	57.17 ± 7.95	0.251 ^a^
Sex	Female	51	37	0.316 ^b^	26	20	0.293 ^b^	11	7	0.426 ^b^	14	10	0.765 ^b^
	Male	80	76		35	40		22	22		23	14	
Lesion location	Supratentorial	128	91	0.001 ^b^	61	50	0.001 ^b^	30	19	0.014 ^b^	37	22	0.074 ^b^
	Infratentorial	3	22		0	10		3	10		0	2	

^a^ Student *t*-test; ^b^ Chi-squared test; ^c^ Mann–Whitney U test.

**Table 2 cancers-13-05793-t002:** Performance comparison of the proposed RFO model with the three neuroradiologists and the MDT-decision voted by three specialists on the two independent testing cohorts. Values in bold indicate the best results.

Models		Independent Testing Cohort 1 (*n* = 62)				Independent Testing Cohort 2 (*n* = 61)			
		AUC	ACC	SEN	SPE	AUC	ACC	SEN	SPE
Top 3 models’ mean		0.916	0.852	0.857	0.843	0.864	0.825	0.708	0.901
Proposed RFO model (radiomics)		0.925	0.855	0.856	0.853	0.859	0.836	0.708	0.919
Proposed RFO model (age + radiomics)		0.922	0.855	0.857	0.853	**0.866**	0.820	0.708	0.892
Proposed RFO model (location + radiomics)		**0.929**	**0.871**	**0.893**	0.853	0.858	0.836	0.708	0.919
Proposed RFO model (age + location + radiomics)		0.927	0.855	0.848	**0.860**	0.865	**0.852**	**0.750**	**0.919**
Neuroradiologists	#1 (3 years experiences)	0.607	0.597	0.576	0.621	0.610	0.607	0.625	0.595
	#2 (5 years experiences)	0.628	0.629	0.546	0.724	0.658	0.656	0.667	0.649
	#3 (15 years experiences)	0.754	0.758	0.758	0.759	0.782	0.770	0.833	0.730
MDT-decision of three specialists		0.722	0.726	0.788	0.655	0.692	0.689	0.708	0.676

**Table 3 cancers-13-05793-t003:** Statistical comparison of the proposed RFO model with three neuroradiologists and the MDT-decision of three specialists on two independent testing cohorts. Values in bold indicate a significant difference.

Models	Independent Testing Cohort 1 (*n* = 62)				Independent Testing Cohort 2 (*n* = 61)			
	AUC	ACC	SEN	SPE	AUC	ACC	SEN	SPE
*p*-value (RFO vs. mean performance of three neuroradiologists)	**0.03**	**0.01**	**0.02**	**0.01**	**0.02**	**0.01**	0.45	**0.02**
*p*-value (RFO vs. MDT-decision of three specialists)	**0.03**	**0.02**	**0.03**	**0.02**	**0.03**	**0.02**	0.44	**0.03**

**Table 4 cancers-13-05793-t004:** Top 10 most frequently selected features of T_1_WI and T_2__FLAIR based on VOI_ET_. The ‘^a^’ and ‘^b^’ represent a Mann–Whitney U test and Independent t test, respectively. The ‘M’ shows the mean of the mean feature value of the GBM group and the mean feature value of the SBM group. The letter of ‘(<M | >M)’ represents the percentage of patients in the GBM and SBM groups with feature value less than or larger than the ‘M’ value. Values in bold indicate these discriminative features with ~70% of the GBM group had larger feature values, while ~70% of the SBM group had smaller values.

Category	Top10 Features	*p*-Value		M		(<M | >M)	
		T_1_WI	T_2__FLAIR	T_1_WI	T_2__FLAIR	T_1_WI	T_2__FLAIR
**Firstorder (*n* = 4)**	90Percentile (1st)	0.016 ^a^	<10^−7, a^	1.01	2.86	GBM (50.82% | 49.18%)	**GBM (27.87% | 72.13%)**
						SBM (66.67% | 33.33%)	**SBM (73.33% | 26.67%)**
	Median (3rd)	0.707 ^a^	<10^−10, b^	0.66	2.17	GBM (44.26% | 55.74%)	**GBM (32.79% | 67.21%)**
						SBM (50.00% | 50.00%)	**SBM (81.67% | 18.33%)**
	Maximum (8th)	<10^−4, a^	<10^−8, a^	1.85	4.02	GBM (55.74% | 44.26%)	**GBM (31.15% | 68.85%)**
						SBM (76.67% | 23.33%)	**SBM (80.00% | 20.00%)**
	Range (10th)	<10^−6, a^	<10^−8, a^	2.03	3.91	GBM (36.07% | 63.93%)	GBM (36.07% | 63.93%)
						SBM (80.00% | 20.00%)	SBM (80.00% | 20.00%)
Shape (*n* = 6)	MinorAxisLength (2nd)	<10^−11, a^	<10^−11, a^	49.30	48.8	GBM (44.26% | 55.74%)	GBM (37.70%| 62.30%)
						SBM (86.67% | 13.33%)	SBM (86.67% | 13.33%)
	Maximum2DDiameterColumn (4th)	<10^−13, a^	<10^−13, a^	62.20	61.60	**GBM (34.43% | 65.57%)**	**GBM (31.15% | 68.85%)**
						**SBM (88.33% | 11.67%)**	**SBM (88.33% | 11.67%)**
	Maximum2DDiameterSlice (5th)	<10^−12, a^	<10^−12, a^	64.37	63.73	GBM (44.26% | 55.74%)	GBM (44.26% | 55.74%)
						SBM (90.00% | 10.00%)	SBM (90.00% | 10.00%)
	Flatness (6th)	<10^−8, a^	<10^−8, a^	0.57	0.57	GBM (49.18% | 50.82%)	GBM (50.82% | 49.18%)
						SBM (8.33% | 91.67%)	SBM (8.33% | 91.67%)
	MajorAxisLength (7th)	<10^−14, a^	<10^−13, a^	61.79	61.05	GBM (42.62% | 57.38%)	GBM (40.98% | 59.02%)
						SBM (91.67% | 8.33%)	SBM (90.00% | 10.00%)
	VoxelVolume (9th)	<10^−9, a^	<10^−9, a^	34001.88	33353.47	GBM (39.34% | 60.66%)	GBM (37.70% | 62.30%)
						SBM (76.67% | 23.33%)	SBM (76.67% | 23.33%)

## Data Availability

Due to privacy or ethical restrictions, the data that support the findings of this study are not publicly available but are available on request from the corresponding author.

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
