# Peer review of "A Multiparametric MR-Based RadioFusionOmics Model with Robust Capabilities of Differentiating Glioblastoma Multiforme from Solitary Brain Metastasis"

_cancers, 2021, doi:10.3390/cancers13225793_

Round 1

Reviewer 1 Report

The desired improvement of the discussion has been made. 

Author Response

We really appreciate the reviewer’s recognition and positive recommendation of publication after minor revisions.

Reviewer 2 Report

The authors describe a multiparametric classification approach to differentiate between GBM and metastatic brain tumors.
The manuscript is comprehensibly written.
- Results could be further discussed: 
e.g. Figure 5 could point out if there is significance in differences between groups. It seems like F2flair is not less accurate than combinations with additional sequences.
- Figure and Table descriptors could be extended for better comprehension.
e.g. Table 4 legend could include above line, indicate what is bold etc.

- Regarding materials and Methods, unfortunately the authors state, that data is not available due to privacy or ethical restrictions.. Maybe they could describe whether to provide parts of the data translated/anonymized and contribute to research data endeavours such as the one from RSNA-ASNR-MICCAI, TCIA, etc. For reproducibility reasons, readers would highly benefit from links to implementation and possibly parts of testing data at least or maybe authors could give links to implementation and state what they think of using the model on open data-sets.

Author Response

We thank Reviewer #2 for this suggestion, which were very helpful for improving our manuscript. We have added the statistical comparison results in Figure 5 in the revised manuscript and Appendix A1 (Table S2). It better showed the statistically meaningful performance superiority obtained by the fusion of T1WI and T2_FLAIR over other feature types, which further demonstrates the validity of our proposed model.

The relevant definitions of bold values and symbols are provided in the legend of Table 4 in the revised manuscript.

We have initiated the process of public sharing our data and  fileded a request at TCIA to upload the data. It may probably take several months to complete the submission process (as described by TCIA) since considerable time is required for data compliance review. On the other hand, we are still waiting for the administrative approval by the hospital’s Ethics Committee before we can publicize  the data.
